# Incorporation of Corrosion Effects into the Life-Cycle Analysis of AW-2017A-T4 Aluminium Alloy under Bending Moment

**DOI:** 10.3390/ma13173681

**Published:** 2020-08-20

**Authors:** Łukasz Blacha, Joanna Małecka, Tadeusz Łagoda

**Affiliations:** 1Department of Mechanics and Machine Design, Opole University of Technology, ul. Mikołajczyka 5, 45-271 Opole, Poland; l.blacha@po.edu.pl; 2Department of Materials Engineering, Opole University of Technology, ul. Mikołajczyka 5, 45-271 Opole, Poland; j.malecka@po.edu.pl

**Keywords:** AW-2017A-T4 aluminium alloy, pre-corrosion, bending fatigue

## Abstract

The paper presents the results of fatigue tests of corroded AW-2017A-T4 aluminium alloy samples subjected to an alternating (symmetrical) bending load. Although there are a number of works describing pre-corrosion fatigue in aluminium alloys, relatively few of them concern bending fatigue effects, in some selected alloys only. Here, the AW-2017A-T4 samples were exposed to electrochemical preliminary corrosion by immersion in an electrolyte, a 3.5% solution of NaCl in water. Several variants of series of samples differing in immersion time were tested. Based on the analysis of the results obtained, Basquin’s fatigue characteristics were developed and compared to the characteristics of the material in its nominal state, which allowed for conclusions on the influence of corrosion effects. The characteristic curves show the susceptibility of the test material to corrosive processes, which results in a decrease in fatigue life along with the increase of pre-corrosion time. The samples with longer immersion duration revealed larger surface losses and widespread corrosion pits.

## 1. Introduction

Structures made of aluminium alloys are widely applied in the construction industry [1,2,3], the aerospace and aviation industry [4,5], the automotive industry [6,7] and military applications [8]. The popularity of this material is mainly due to its relatively low density, good strength-to-mass ratio, high corrosion resistance, and good ability to be welded, including with other materials [9,10,11]. These properties make it possible to reduce operating costs: low weight means low energy consumption. In the same way, maintenance costs can also be reduced: corrosion resistance higher than that of ferrous metals allows to plan less maintenance effort. The AW-2017A-T4 alloy is used in the construction of machines and structures as a material for construction elements of aircrafts, vehicles, and military equipment [12,13].

Corrosion processes are the cause of degradation of the material. They arise as a result of the destruction processes and lead to a decrease in strength and operational parameters (i.e., fatigue life). The initiation of such a process has its origin in the chemical or electrochemical environmental impact. In electrochemical corrosion, a typical structure damage is initiated by a corrosion pit, small hole created in a process of localised corrosion. The formation of possible corrosion pits in the material will result in a loss of strength due to a source of stress concentration. Electrochemical corrosion occurs in a corrosive environment, such as an electrolyte. In such an environment, local depassivation occurs on a small area of the metal which becomes an anode while the nearby area behaves as a cathode. An example of an electrolyte is a solution of water with a dissolved substance accelerating corrosion. Increased resistance of aluminium alloys to electrochemical corrosion is due to the phenomenon of passivation, i.e., the formation of a passive layer on the external surface of the material, which prevents the formation of corrosive cells. The retention of such a state depends on the electrode potential. Therefore, the comparative analysis of operational parameters in atmospheric conditions and after a period of functioning in a corrosive environment (in an electrolytic conductor) is important to analyze. One of the operational parameters is fatigue life, which is characterized by resistance to the accumulation of damage in the material caused by the cyclic load.

There are a number of works concerning the development of fatigue cracks in various aluminium alloys subjected to corrosion [14,15,16], but they usually concern one particular type of environment with strictly defined physicochemical parameters, and the analysed parameter is only the crack development rate, without direct reference to fatigue life. There is also a large number of works describing experimental fatigue strength, but mainly in tension or compression conditions [17,18,19,20,21]. Relatively few of them concern bending or multiaxial conditions, and these are mainly in specialised applications. (e.g., [22]). The knowledge on material resistance to bending fatigue is important in understanding the principles of the effect of corrosion on fatigue properties [23,24]. It becomes especially important in the case of long-span and truss structures [25], axles and shafts [26], airframes [27].

The aim of this work was to explore experimentally life assessment aspects of bending fatigue in the structure–fluid interaction between an aluminium structure and corrosive fluid. The material tested was aluminium alloy AW-2017A-T4 after various periods of exposure to 3.5 wt% NaCl solution in water, i.e., conditions favourable to electrochemical corrosion recognised by the ASTM D1141-98. The obtained data will therefore provide information about the scale of material degradation within a wide range of load history and would be useful in modelling the load–life interaction and the corresponding mechanisms of degradation due to bending fatigue, as the current state of the art still lacks information concerning this topic.

## 2. Experimental Research

The purpose of the tests was to determine the fatigue life in alternating bending conditions. The characteristics of the material tested are presented further in the article.

### 2.1. Material and Samples

The samples were made of the AW-2017A aluminium alloy in a T4 state. Due to the presence of copper in the chemical composition, among other alloys it is classified as difficult to weld and moderately corrosion-resistant [28,29]. The chemical composition and static strength properties of the material are presented in Figure 1 and Table 1, respectively. The material was delivered in the form of a bar. The samples had the geometry of a diabolo specimen (Figure 2). They were cut by saw and then manufactured in the processes of turning and milling. Surface roughness *Ra* before the corrosion was 1.25.

The samples were subjected to corrosion over several days and then used for fatigue testing. The examination of the decohesion surface of the material tested and the evaluation of corrosion damage were carried out with the use of scanning microscope JEOL JSM-840. Secondary electron detection (SE) and backscattered electron detection (BSE) were used to obtain fracture images.

### 2.2. Corrosion Process

The corrosion process of the samples was initiated by immersion in an electrolyte, a 3.5 wt% NaCl solution in water, for 7, 14 or 28 days (depending on the test series). A 3.5% salinity was chosen as a reproducible solution for tests on corrosion, recognised by the ASTM D1141-98 standard for the preparation of saline water. The tests were conducted on two series of samples prepared in this way. Samples from the first series were immersed in a solution in tap water; samples from the second series were immersed in a solution in demineralised water. Both series were prepared in several variants differing in the time of immersion in the electrolyte, lasting 7 or 14 days, with an additional variant of 28 days in the case of series in which demineralised water was the solvent (Table 2). Such long immersion periods were chosen in order to fully test the influence of pre-corrosion time on fatigue life, since many tests are accelerated tests (such as salt spray).

The samples of a given series were immersed at the same time and in the same container from which all samples of the variant were removed after an appropriate period of time (i.e., after 7, 14 or 28 days, depending on the variant), then washed with demineralised water, and dried.

The controlled corrosion process took place at room temperature. Its dynamics were determined as dependent on temperature and the properties of the electrolyte, together with the chemical composition of the water. The chemical composition of the water, considered in preparation of the solution used for the preparation of the first series of samples, was assessed using a water quality strip tester (Table 3).

The results obtained also made it possible to test the concentration of hydrogen ions, which were determined to be 6.0 on the pH scale, and to have a hardness of 180 ppm.

### 2.3. Fatigue Tests

The tests were carried out at the monitored bending moment at a frequency of 28.8 Hz. The samples were subjected to constant-amplitude bending loads where asymmetry coefficient *R* = −1. All tests were carried out on the MZGS-100 test stand [30] made at the Opole University of Technology, which enables fatigue testing in combined conditions of bending and torsion load (Figure 3).

The load was applied on the test bench by setting the clamping lever on one side of the sample in a vibrating motion. The load combination was realized by an appropriate angular offset of the column holding the other side of the sample. The tests were carried out with zero offset.

The failure criterion was assumed to be a 15% decrease in bending moment, appearing always at the very end of test. The corresponding number of cycles was assumed as the durability, *N_f_*. The stress on the sample surface was determined according to the linear-elastic body model under the assumption of flat bending (Equation (1)).

## 3. Results

The results of fatigue tests are shown in Table 4, Table 5, Table 6, Table 7 and Table 8, where *M_b,a_* is bending moment amplitude, and *σ_a_* is stress amplitude, calculated as follows:(1)$$\sigma_{a}\;=\;\frac{M_{b,a}}{W}\;=\;\frac{32M_{b,a}}{\pi d^{3}}$$
where *W* = the bending resistance moment about the neutral axis, *d* = the diameter of the sample on the test section—here, 10.2 mm.

Pictures of the fatigue fracture of the samples are shown in Figure 4 and Figure 5. The clearly visible origin and the lack of front lines indicate uniform propagation of the fatigue crack. Microscopic images of crack origin showing the pit are shown in Figure 6 and Figure 7.

Fractographic observation of the tested material, both for the first and second series of tests, shown a typical ductile fracture (Figure 8, Figure 9, Figure 10, Figure 11, Figure 12, Figure 13 and Figure 14). Developed crack area was observed within the specimen axis. The pattern of cracking depends on the overall structural phenomena occurring during the strain of the material, but it is also important to take into account the chemical activity of the environment in which the material works, as corrosion promotes the nucleation and development of cracks in structural components.

Figure 8, Figure 9, Figure 10 and Figure 11 show fracture images after 7 and 14 days of testing in tap water, respectively, where the zone of the ductile fracture of the tested alloy is visible.

Figure 12, Figure 13 and Figure 14 shows fractographic fracture images after 7, 14 and 28 days of test conducted in demineralised water.

The observations carried out with the scanning electron microscope (SEM) allowed to conclude that the tested alloy is subjected to local pitting corrosion as a result of the break of passive layer. This phenomenon occurred regardless of the immersion duration and corrosive environment. However, more pits were formed on the surface of the alloy being longer exposed to the corrosive environment (Figure 15 and Figure 16).

This phenomenon indicates a significant influence of chloride ion concentration in the corrosive medium on the formation of pits. Although pits are formed evenly over the entire exposed area of the sample during both tests: in tap and demineralised water, the damage increases along with the immersion duration. Figure 17 show pictures of the interior of pits created during corrosion tests. They indicate that the surface is significantly etched. The pitting starts on the surface of the material, but also quickly progresses inwards. Very often the rest of the metal on the surface remains intact. Such a phenomenon usually occurs in materials that are subject to self-passivation, such as stainless steel or aluminium alloys. Usually, pits are first nucleated on the metal surface and then develop and propagate inwards, hence the visible pits at material fractures. Pit nucleation occurs at the weakest points of the passive layer: in places of mechanical damage, near some non-metallic inclusions or at grain boundaries. Probably, the nucleation is preceded by adsorption of aggressive ions, especially Cl, on the surface. Then the ions penetrate the passive layer by migration or penetration.

The observed fatigue strength allowed to formulate the S-N fatigue characteristics in a double logarithmic system (according to ASTM [31]):(2)$$\log N_{f}\;=\;B-m\log\sigma_{a},\;$$
where *N_f_* = durability, the number of cycles to failure, -; *B* = material parameter, -; *m* = slope, -; *σ_a_* = elastic stress amplitude, MPa.

Fatigue characteristics were developed separately for each variant and test series. In order to determine the values of parameters *B* and *m* in Equation (2), a linear regression model was used. The quality of approximation was assessed by means of correlation coefficient *ρ*. The values obtained are shown in Table 9. The processed data required to reproduce these findings can be found at [32].

The obtained results were compared to the results of the tests of samples in their nominal state, i.e., not subjected by a corrosive environment, described, among others, in [33,34,35]. The formulated characteristics were compared with the nominal characteristics determined by Kurek et al. [33], with Equation (2) parameters *B* = 21.71 and *m* = 7.03. The comparison of fatigue characteristics of both test series (Table 4, Table 5, Table 6, Table 7 and Table 8) with nominal characteristics is shown in Figure 18 and Figure 19, respectively.

## 4. Discussion

The paper described two series of fatigue tests of samples made of aluminium alloy AW-2017A under cyclic bending load, exposed to electrochemical corrosion caused by immersion in an electrolyte, a 3.5 wt% solution of NaCl in water. The first series of tests concerned a solution in tap water, and the second series concerned a solution in demineralised water. In each series, fatigue fractures indicated uniform crack propagation. The obtained test results feature Pearson correlation coefficient values between −0.88 and −0.90 in case of tap water solution and between −0.88 and −0.93 in case of demineralised water solution. The comparison showed a decrease in fatigue strength in each case except the 28 days of pre-corrosion serie. In this case it was slightly higher than the 7 and 14 days series, but it has to be mentioned that for this solution all S-N curves are very close to each other.

The microscopic observations show that all samples of the tested alloy have undergone pitting corrosion. The smallest number of pits was formed on the sample after the shortest immersion duration (7 days, Figure 15a and Figure 16a), more pits were visible on the samples immersed for 14 days (Figure 15b and Figure 16b), which is consistent with the results obtained from fatigue tests. The observations indicate a disruption of the anode layer and the occurrence of micro cracks which transform into corrosion. In addition, it is not possible to create a passive layer on the surface of the material and for this reason it is easier to initiate pitting on its surface. Larger corrosion damage probably results from the extension of the time of exposure to aggressive corrosive environment, which simultaneously translates into shorter fatigue life of the material. The results have shown that after a longer period of exposure to NaCl solution pits are formed on the surface of the samples with higher density.

From the additional observations it can be seen that the propagation of a crack was accompanied by intense plastic strain and accumulation of deformation strain energy in the material. Plastic deformation of the material was initiated in the surface layer in grains with the crystallographic orientation in the easy glide direction and near the stress concentrators (i.e., in weakened parts of the material). The process of ductile cracking is associated with high plastic strain, so it requires a significant energy expenditure for crack propagation. The topography of the developed crack area consists of a large collection of pits (craters) of various sizes and shapes, which can be viewed in Figure 9a, Figure 11a, Figure 12a, Figure 13a and Figure 14a. Corrosion pits were observed also at the crack origin. In the case of tests carried out in tap water and in demineralised water it was observed that with increasing exposure time to the corrosion environment, the corrosive pits became more widespread (Figure 9a, Figure 11a, Figure 12a, Figure 13a and Figure 14a), which caused geometric irregularities in the material. Such a phenomenon leads to local plastic deformation of the material. It points on plastic fatigue cracking, which is indicated by the ductile fractures of the analysed samples (Figure 9b,c, Figure 11b,c, Figure 12b,c, Figure 13b,c and Figure 14b,c).

In the case of the first series of tests, a clear influence of the immersion duration can be observed. This can be explained through the fact that with the increase of corrosion pits fatigue crack initiation life becomes shorter, although this trend decreases in the high cycle fatigue region. These results can be interpreted as evidence of the importance of the joint effect of stress and corrosion processes on high cycle fatigue behaviour. Globally, the results showed a decrease in the conventional fatigue limit (i.e., stress at 1 × 10^7^ cycles) and a slight change in the slope of the S-N diagram, which translates into a higher susceptibility to failure and a decrease in the influence of bending stress on fatigue life, with an increase in the time of corrosive processes. The above observation indicates a lesser influence of material properties on its fatigue strength at low stress values and high number of cycles. In this region curves for both immersion durations are very close to each other and the obtained life would be close for both of them.

The reason for the differences in fatigue life can be attributed to the presence of plastic strain, which considerably influences fatigue processes in aluminium alloys [36,37], as was also shown in fractographic tests. Yang et al., through fatigue tests of different alloys, have proved that plastic properties strongly influence fatigue lives [36]. At higher stress amplitudes, the presence of corrosion pits lead to local plastic strain and the coalescence of flaws. The higher the amplitudes, the deeper the cavity is. The topography of the obtained fracture planes exhibited differences between samples of different immersion times. From the analysis of the two fracture planes in Figure 4, it can be seen that the crack originated always at the surface and that the rapid fracture region has slightly decreased with the elongation of immersion time. The fractography evidenced the typical ductile behavior of the fatigue fracture with the increased rate of plastic strains in the case of the samples with longer immersion times. This further indicates the existence of local stress concentrations of a greater magnitude than in the case of shorter immersion times. This can be attributed to the notch effect of a geometrical origin, such as pitting. The pitting mechanism promotes intergranular corrosion attack, which can result in geometrical discontinuities. Corrosion pits initiated the formation of channels linking the corroded surface and uncorroded material interior, which caused discontinuities and the expansion of corrosion processes.

In the high cycle fatigue, the results are most likely affected by the notch effect of material origin. Due to the ongoing electrochemical reactions, corrosion processes are also responsible for the production of hydrogen on the material surface. The production of hydrogen can evolve and develop from the surface into the interior of the material, along with the crack tip. Such behaviour causes local embrittlement, responsible for variation in local fracture toughness [38,39]. This is typical at low strain rates [39].

In the case of the second series of tests, where demineralised water was the solvent, the observed decrease was almost independent of the length of immersion time. This observation was confirmed during additional tests consisting of 28 days’ immersion in solution. The difference in durability between the samples from both series is most likely caused by different corrosion dynamics, due to the properties of such an electrolyte, determined by low solvent conductivity.

## 5. Conclusions

The paper describes bending fatigue tests of samples pre-corroded in a 3.5% wt. solution of NaCl in tap and demineralised water. From the results, the following conclusions can be drawn:(1)The results from both series of tests showed a decrease in fatigue life due to the susceptibility of the tested material to pitting corrosion processes and the plastic strain mechanisms;(2)The results of tests of samples from the series immersed in the solution in tap water showed a decrease in stress at 1 × 10^7^ cycles (i.e., the conventional fatigue limit), nearly the same in case of 7 and 14 days of immersion duration, although a slope different between both S-N diagrams can be observed. Different slopes which translates into a higher susceptibility to failure and a decrease in the influence of bending stress on fatigue life with an increase in pre-corrosion time;(3)In the case of the samples from the series immersed in the solution in demineralised water, only a slight impact of the length of time spent in the corrosive medium on fatigue life was observed. The observation on samples after 7 and 14 days in solution was confirmed by the results of additional fatigue tests, on samples after 28 days in solution. The difference in durability between the samples from both series is most probably caused by different corrosion dynamics resulting from the properties of such an electrolyte, determined by low solvent conductivity;(4)fatigue fractures indicate uniform crack propagation;(5)the obtained test results feature Pearson correlation coefficient value not less than −0.88.

## Figures and Tables

**Figure 1 materials-13-03681-f001:**
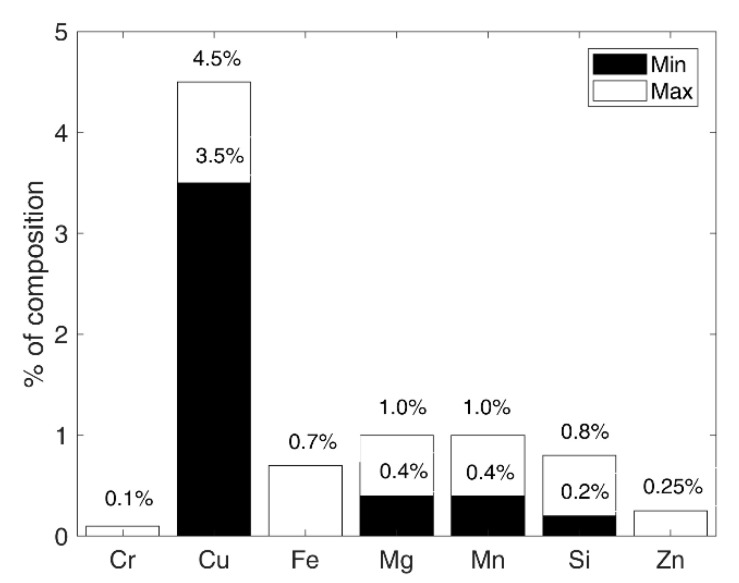
Chemical composition of the test material according to EN 573-3 standard.

**Figure 2 materials-13-03681-f002:**
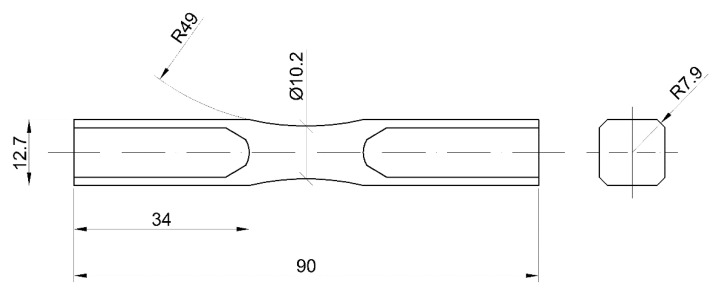
Geometry of samples used for fatigue testing (mm).

**Figure 3 materials-13-03681-f003:**
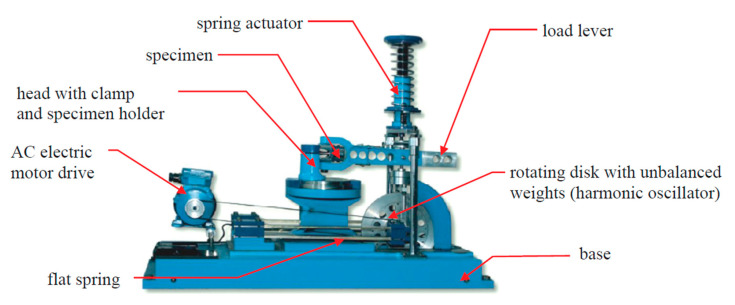
The overall view of the MZGS-100 test stand [30].

**Figure 4 materials-13-03681-f004:**
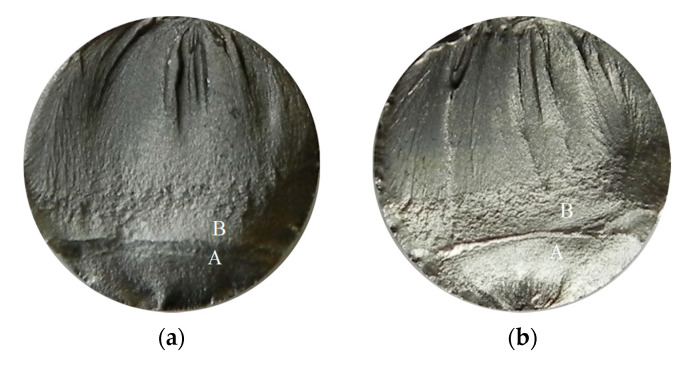
Fatigue fracture of exemplary samples from a series subjected to the solution in tap water, submitted to stress amplitude *σ_a_* = 162 MPa: (**a**) 7 days of immersion (Sample No. 3, Table 4); (**b**) 14 days of immersion (Sample No. 15, Table 5). Cracks originated from the bottom of the specimen, where: A—crack growth region, B—fast fracture zone.

**Figure 5 materials-13-03681-f005:**
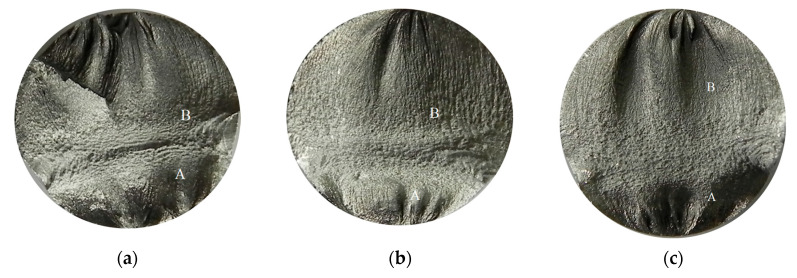
Fatigue fracture of an exemplary samples from a series subjected to the solution in demineralised water, submitted to stress amplitude *σ_a_* within a range from 217 to 227 MPa: (**a**) 7 days of immersion (Sample No. 2, Table 6); (**b**) 14 days of immersion (Sample No. 6, Table 7); (**c**) 28 days of immersion (Sample No. 4, Table 8). Cracks originated at the surface, from the bottom of the specimen, where: A—crack growth region, B—fast fracture zone.

**Figure 6 materials-13-03681-f006:**
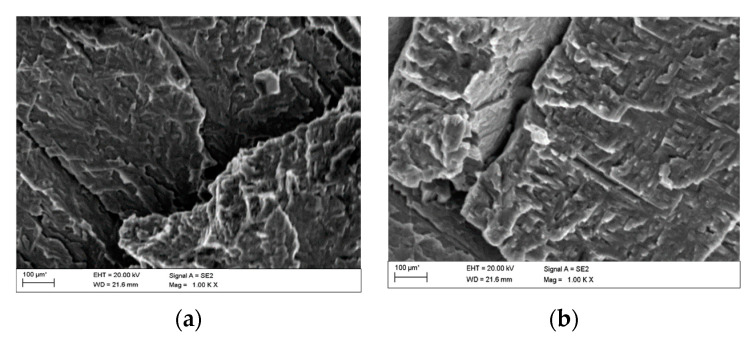
Microscopic images of crack origin, sample from a series subjected to the solution in tap water: (**a**) sample in Figure 4a, (**b**) sample in Figure 4b.

**Figure 7 materials-13-03681-f007:**
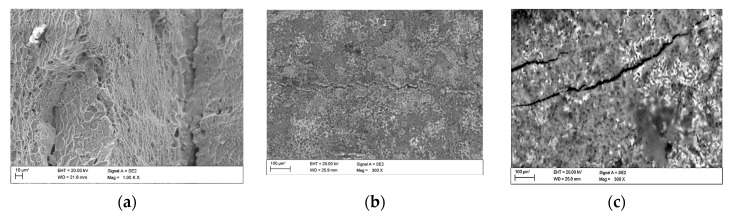
Microscopic images of crack origin, sample from a series subjected to the solution in tap water: (**a**) sample in Figure 5a, (**b**) sample in Figure 5b, (**c**) sample in Figure 5c.

**Figure 8 materials-13-03681-f008:**
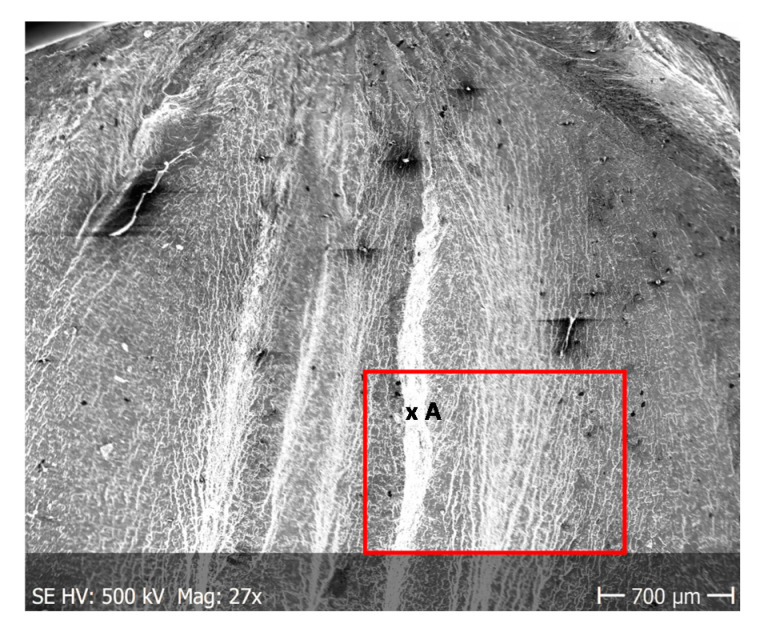
Fracture images after 7 days of testing in tap water (sample shown in Figure 4a).

**Figure 9 materials-13-03681-f009:**
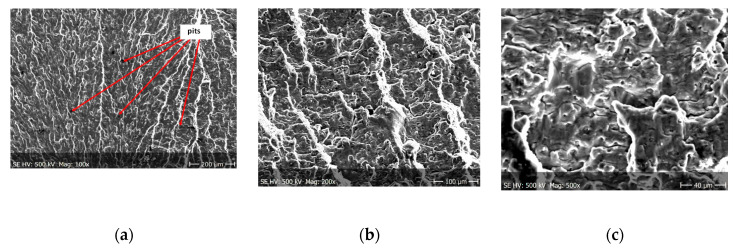
(**a**–**c**) Fracture images within area A (Figure 8).

**Figure 10 materials-13-03681-f010:**
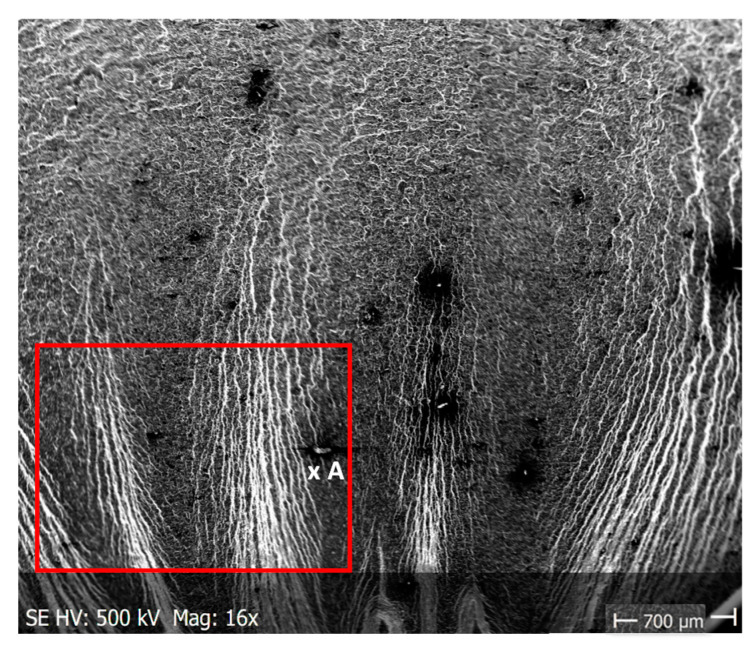
Fracture images after 14 days of testing in tap water (sample shown in Figure 4b).

**Figure 11 materials-13-03681-f011:**
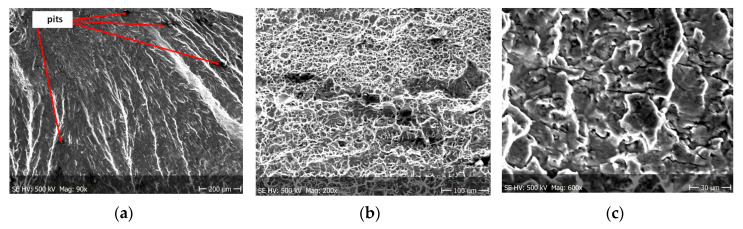
(**a**–**c**) Fracture images within area A (Figure 10).

**Figure 12 materials-13-03681-f012:**
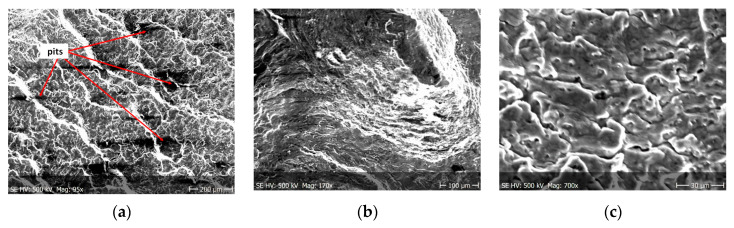
(**a**–**c**) Fracture images after 7 days in demineralised water (images taken at fracture surface of sample shown in Figure 5a).

**Figure 13 materials-13-03681-f013:**
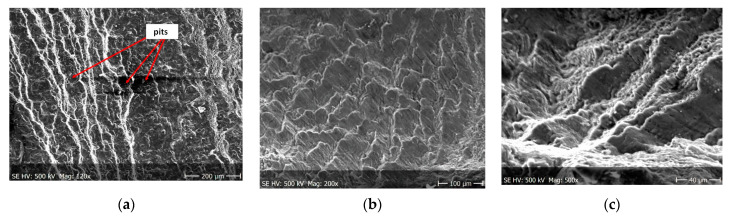
(**a**–**c**) Fracture images after 14 days in demineralised water (images taken at fracture surface of sample shown in Figure 5b).

**Figure 14 materials-13-03681-f014:**
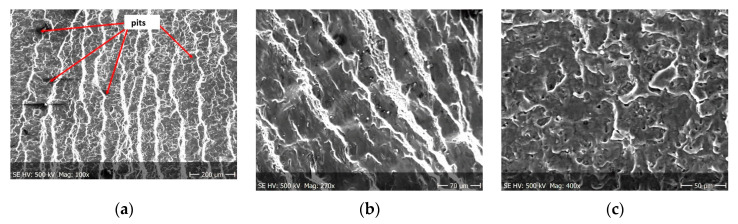
(**a**–**c**) Fracture images after 28 days in demineralised water (images taken at fracture surface of sample shown in Figure 5c).

**Figure 15 materials-13-03681-f015:**
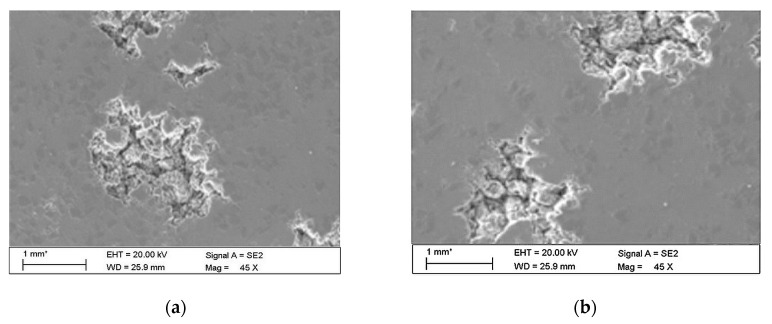
Surface of tested aluminium alloy after 7 (**a**) and 14 (**b**) days of testing in tap water. Visible changes in form of pits.

**Figure 16 materials-13-03681-f016:**
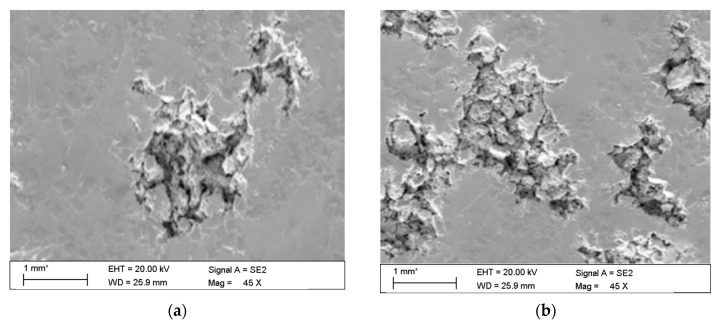
Surface of tested aluminium alloy after 7 (**a**) and 14 (**b**) days in demineralised water. Visible changes in form of pits.

**Figure 17 materials-13-03681-f017:**
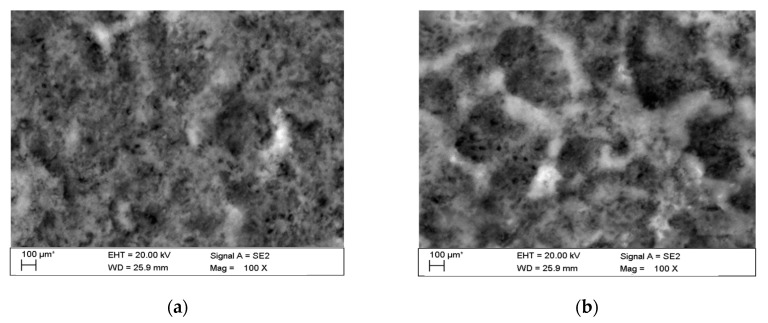
Interior of pits created during corrosion tests: (**a**) in tap water, (**b**) in demineralised water.

**Figure 18 materials-13-03681-f018:**
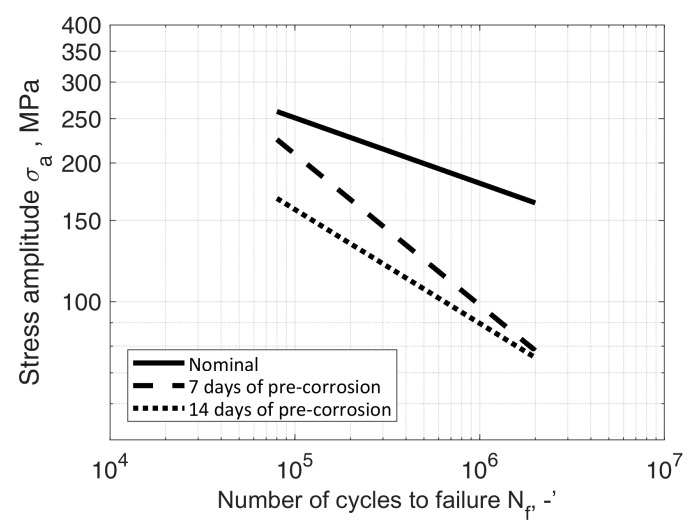
S-N data of AW-2017A-T4 material in nominal condition (room temperature, in air) and samples immersed in tap water solution for 7 and 14 days prior to bending fatigue tests.

**Figure 19 materials-13-03681-f019:**
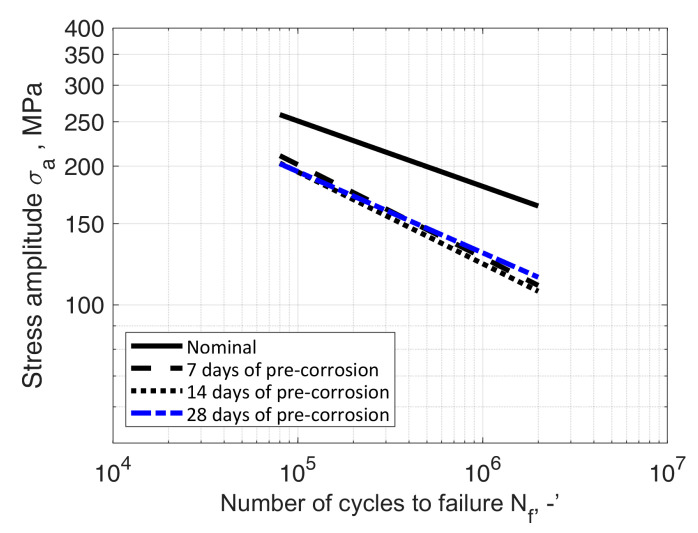
S-N data of AW-2017A-T4 material in nominal condition (room temperature, in air) and samples immersed in demineralised water solution for 7, 14 and 28 days prior to bending fatigue tests.

**Table 1 materials-13-03681-t001:** Selected strength properties of the test material according to EN 485-2+A1:2018-12 standard.

UTS	*YS_2%_*	A
min, *MPa*		min, %
350	260	13

**Table 2 materials-13-03681-t002:** Test series.

Sample Series	Electrolyte	Immersion Duration
First series	3.5 wt% NaCl solution in tap water	7 and 14 days
Second series	3.5 wt% NaCl solution in demineralised water	7, 14 and 28 days

**Table 3 materials-13-03681-t003:** Chemical composition of water considered in preparing the first series of samples.

Fe	Cl	Nitrates	Nitrites
*Ppm*			
0.1	0.1	25.0	1.0

**Table 4 materials-13-03681-t004:** Results of fatigue tests of samples after 7 days of immersion in a 3.5% NaCl solution in tap water.

No.	*M_b,a_*	*σ_a_*	*N_f_*
	Nm	MPa	-
1	16.4	162.14	297,798
2	16.6	164.11	271,494
3	16.4	162.14	125,644
4	11.0	108.75	852,785
5	10.6	104.79	964,988
6	10.8	106.77	485,289
7	10.9	107.76	605,299
8	10.5	103.81	1,208,290

**Table 5 materials-13-03681-t005:** Results of fatigue tests of samples after 14 days of immersion in a 3.5% NaCl solution in tap water.

No.	*M_b,a_*	*σ_a_*	*N_f_*
	Nm	MPa	-
1	12.5	123.58	276,742
2	15.2	150.27	104,184
3	15.7	155.22	883,96
4	12.6	124.57	178,284
5	11.1	109.74	328,443
6	15.7	155.22	126,142
7	12.6	124.57	263,049
8	11.4	112.70	641,145
9	11.4	112.70	446,843
10	10.4	102.82	619,696
11	10.4	102.82	1,100,137
12	11.5	113.69	283,144
13	11.1	109.74	317,796
14	11.3	111.72	284,989
15	16.4	162.14	88,547
16	10.9	107.76	410,442
17	11.1	109.74	620,846
18	16.5	163.12	144,449

**Table 6 materials-13-03681-t006:** Results of fatigue tests on samples after 7 days in a 3.5% NaCl solution in demineralised water.

No.	*M_b,a_*	*σ_a_*	*N_f_*
	Nm	MPa	-
1	22.7	224.42	53,198
2	23.0	227.39	56,089
3	23.0	228.37	56,387
4	15.4	152.25	374,987
5	15.7	155.22	217,172
6	15.3	151.26	529,011
7	15.3	151.26	850,634
8	19.2	189.82	185,861
9	18.9	186.85	136,600
10	18.3	180.92	166,471
11	16.0	158.18	191,783

**Table 7 materials-13-03681-t007:** Results of fatigue tests on samples after 14 days in a 3.5% NaCl solution in demineralised water.

No.	*M_b,a_*	*σ_a_*	*N_f_*
	Nm	MPa	-
1	14.0	138.41	1,486,798
2	13.8	136.43	610,621
3	13.8	136.43	321,025
4	14.1	139.40	199,693
5	22.2	219.48	77,134
6	22.4	221.45	52,587
7	22.0	217.50	45,241
8	18.7	184.87	99,541
9	14.3	141.37	988,046

**Table 8 materials-13-03681-t008:** Results of fatigue tests of samples after 7 days in a 3.5% NaCl solution in demineralised water.

No.	*M_b,a_*	*σ_a_*	*N_f_*
	Nm	MPa	-
1	14.0	138.41	1,945,988
2	14.2	140.39	587,437
3	12.3	121.60	1,839,387
4	22.0	217.50	66,984
5	13.5	133.47	734,042
6	12.8	126.55	768,036
7	20.8	205.64	65,542
8	20.7	204.65	80,335
9	15.6	154.23	165,833

**Table 9 materials-13-03681-t009:** Fitting parameters.

Solvent	Pre-Corrosion Time	B, -	m, -	*ρ*, -
Tap water	7 days	12.06	3.04	−0.88
	14 days	13.88	4.03	−0.90
Demineralised water	7 days	16.42	4.95	−0.92
	14 days	16.47	5.01	−0.88
	28 days	17.96	5.66	−0.93

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
