# Peer review of "Incorporation of Corrosion Effects into the Life-Cycle Analysis of AW-2017A-T4 Aluminium Alloy under Bending Moment"

_materials, 2020, doi:10.3390/ma13173681_

Round 1

Reviewer 1 Report

I appreciate the efforts of the authors. When i´m talking about defect characterization, I mean that once the defects are caused in the material (with certain size and shape), their geometry should be characterized. More aggressive environments would cause larger defects and shorter fatigue lives. Such relation should be analysed, but it is not. That´s why my concerns remain.

Author Response

Dear Reviewer.

Thank you for all the remarks concerning the article. Detailed information on the changes can be found below. It has been re-edited following also the remarks by other reviewers. We sincerely hope that now it is written in a convenient and proper manner.

Comment:

“When i´m talking about defect characterization, I mean that once the defects are caused in the material (with certain size and shape), their geometry should be characterized. More aggressive environments would cause larger defects and shorter fatigue lives. Such relation should be analysed, but it is not. That´s why my concerns remain.”

Response:

New lines have been added to the Results and the Discussion sections, as well as images from microscopic observations (Figures 8, 9a, 10, 11a-14a, 15-17). Due to the amount of data and changes made, the text and Figures are not pasted here, they can be found directly in the manuscript.

Yours sincerely,

                                                                                                                                                                                                                                     Łukasz BLACHA,

                                                                                        Joanna MAŁECKA,    

                                                                                        Tadeusz ŁAGODA

Reviewer 2 Report

The manuscript presents a study about incorporation of corrosion effects into the life-cycle analysis of a bent aluminium 2017-T4 alloy. This is an interesting work and within the scope of materials. However, this paper seems lack of in-depth discussion. I can not recommend this manuscript for publishing in materials. There are some major points which need to be considered by the authors for improving this manuscript.

  1. Line 2 aluminium should be aluminum.
  2. It is well known the fatigue life is sensitive to the condition and configuration of the surface of metallic substrate. The density, depth and morphology of pitting corrosion for different corrosion process should be addressed and compared in the manuscript. In addition, these observations should be connected to the fatigue results.
  3. There are more descriptions of the results but fewer in-depth discussions in Results and Discussion section.
  4. Does fatigue crack origin change with respect to the different pre-corrosion process? Fatigue crack origin should be addressed in the fatigue fracture observation.
  5. Conclusion section is missing.

Author Response

Dear Reviewer.

Thank you for all the remarks concerning the article. Detailed information on the changes can be found below. It has been re-edited following also the remarks by other reviewers. We sincerely hope that now it is written in a convenient and proper manner.

Comment 1:

“Line 2 aluminium should be aluminum.”

Response:

The manuscript is written in UK English, therefore in the title there is “aluminium”  instead of “aluminum”.

Comment 2:

“It is well known the fatigue life is sensitive to the condition and configuration of the surface of metallic substrate. The density, depth and morphology of pitting corrosion for different corrosion process should be addressed and compared in the manuscript. In addition, these observations should be connected to the fatigue results.”

Response:

New lines have been added to the Results and the Discussion sections, as well as images from microscopic observations (Figures 8, 9a, 10, 11a-14a, 15-17). Due to the amount of data and changes made, the text and Figures are not pasted here, they can be found directly in the manuscript.

Comment 3:

“There are more descriptions of the results but fewer in-depth discussions in Results and Discussion section.”

Response:

Results and discussion section in now re-edited. There are new analysis, some of the sentences from previous sections were moved into this section as being more appropriate. The changes are not listed here due to their amount. They are visible within the manuscript (the “track changes” option).

Comment 4:

“Does fatigue crack origin change with respect to the different pre-corrosion process?

Fatigue crack origin should be addressed in the fatigue fracture observation.”

Response:

Crack origin was always at the surface of the sample. The fatigue fracture Figures are now re-edited in order to present crack zones, as well as the corresponding captions. Also, Figures with a close up view on crack origin are now a part of the manuscript (Figures 6 and 7). In view of those figures it can be seen that the crack was initiated from corrosion pits.

Comment 5:

“Conclusion section is missing.”

Response:

A new section is now added.

Yours sincerely,

                                                                                                                                                                                                                                     Łukasz BLACHA,

                                                                                        Joanna MAŁECKA,    

                                                                                        Tadeusz ŁAGODA

Reviewer 3 Report

In this paper, the results of fatigue tests on two series of corroded AW-2017A-T4 aluminium alloy samples under bending loading are presented. The S-N data of corroded samples in two solutions are the main contribution that can be useful for fatigue life prediction in some applications. Thus, I recommend the paper to be published in "Materials Journal" after a major revision. The following clarifications and modifications should be provided in the paper:

  1. The title should be modified to ‘Incorporation of….. analysis of AW-2017A-T4 aluminium alloy under bending moment’
  2. Please use ‘series’ instead of ‘ batches’ in the paper.
  3. Line 17 to 19 should be modified to represent obtained results.
  4. Line 23, this parameter is not used in the paper.
  5. Line 24, delete ‘elastic’ and use ‘applied’ instead.
  6. Line 26, use ‘Minimum’ instead of ‘minimal’.
  7. Line 27, use ‘Maximum’ instead of maximal’.
  8. Line 31, use ‘Bending Moment amplitude’ instead of ‘amplitude of bending moment’
  9. Delete ‘, -‘ from the end of all symbols list
  10. Line 35, modify ‘offset yield stress’ to ‘Offset Yield Strength’ and the symbol to YS2%
  11. Line 36, modify ‘Rm’ to ‘UTS’
  12. Line 40, specify ‘others’
  13. The English of the introduction should be improved. It is not easy to read and clear.
  14. More references should be added to the introduction to identify the importance of the problem investigated in this issue and also to highlight the gap in the literature.
  15. Line 42 and 43, modify ‘ wide possibilities of joining in processes such as melting (welding) or friction stir welding (FSW, including with other materials)’ to ‘weldability’.
  16. Line 43 to 45, Sentence ‘These properties make it possible to reduce operating expenses, such as operating costs (low weight of the elements) and maintenance costs (corrosion resistance higher than that of ferrous metals).’ should be modified, it is not easy to read. Operating expense is the same as operating cost!
  17. Line 46, sentence ‘Corrosion processes are the cause of degradation of the material as a result of the destruction processes that lead to a decrease in strength and operational parameters.’ should be modified, it is not good in English and ‘operational parameter’ should be specified.
  18. Line 47, corrosion pit should be defined first.
  19. Line 50, should be modified. The electrolyte is one of the necessary elements for any corrosion cell. The presence of cathodes and anodes are important too.
  20. Line 56, modify ‘is interesting’ to a scientific term.
  21. Line 58, modify ‘time-variable’ to ‘cyclic’.
  22. Line 60, ‘for example’ should be deleted.
  23. Line 64, e.g. should be deleted.
  24. Line 64 and 65 should be improved in writing.
  25. It is not clear why bending fatigue is investigated. Specify some real-life application for this material that is under bending fatigue. The same for used corrosive environment. Why is this corrosive environment chosen? Is it replicating any in-service condition? These all should be described in the introduction.
  26. Line 75 and 76 should be moved to introduction.
  27. Line 80 says this alloy is difficult to weld whereas in line 42 and 42 the opposite is said! The same for being corrosion resistant.
  28. Which specific standard was used to design the samples?
  29. Which surface preparation process was carried after cutting the samples? What was the sample surface roughness before and after corrosion?
  30. The methodology of corroding the samples should be described in detail. Specify any electrodes used, voltage, etc. Add images of the corrosion cell with the samples.
  31. Add image of fatigue test machine with the sample inside it.
  32. The number of samples, duration of the corrosion process and applied stress should be specified here.
  33. Line 90, [EN 573-3] is a reference?
  34. Line 91, [EN 485-2+A1:2018-12] is a reference?
  35. Line 93, add a unit to the caption (mm)
  36. Line 96, specify ‘several days’
  37. Line 99, clarify whether you mean tap water by ‘running water’ or you mean electrolyte was running in the cell through an inlet/outlet.
  38. State the reason for choosing two types of water.
  39. Clarify why 7, 14 and 28 days were chosen.
  40. It is not clear which samples, how many days. Add a table and detail number of days, sample series and type of electrolyte.
  41. Line 104 to 106, do you mean the same solution was used for all samples and it was not refreshed anytime you put new sample?? Clarify this. Fresh solution should have been used for each sample series.
  42. What was room temperature?
  43. Table 2, add chemical composition of water for the first batch of samples to this table. It should be added to enable the reader to compare the two solutions.
  44. Was fatigue tests load controlled or strain-controlled?
  45. Section 2.3. add a figure of the test set up and test bench with the sample inside the fatigue test machine
  46. Line 120, was your fatigue tests rotating bending? Specify what kind of bending load was applied and how.
  47. Line 124, assumption of 15% needs to be supported by references.
  48. Line 126 and 127, add the equation.
  49. Line 129 and 130, delete ‘=’ and use ‘is’
  50. Figure 3: Tables should be introduced before this figure and then use the sample numbers in the caption. Show the crack origin and crack growth region and fast fracture zones on the images by arrows and brackets. State in the cation which microscope was used and how many days they were immersed.
  51. Figure 4: same comment as Figure 3.
  52. Line 143 to 149, clarify how does corrosion promoted fatigue crack in this case.
  53. What is the difference of fig 3 and 4?
  54. Figure 5, which part of cross-section is these image from? add another image from cross-section to fig 5 and show where images are taken from by red boxes.
  55. Same comment for Figure 6.
  56. Line 161, clarify what made you result in the plastic strain in the images.
  57. Line 166, Grammar mistake ‘it was be observed that’
  58. Line 167, show pits with an arrow on fig 5a to 9a.  ‘÷’ modify to ‘-‘
  59. Line 167, change ‘corrosive’ to ‘corrosion’
  60. Line 167, what was the corrosion pit depth? Clarify.
  61. Have you observed corrosion pits at the crack origin? Add microscopic images of crack origin showing the pit.
  62. Line 169, English should be improved.
  63. Line 170, modify ‘÷’ to ‘to’
  64. Line 165, clarify which pits you mean on the images.
  65. Line 184, delete ‘one’
  66. Line 201, use the number of tables here instead of the figures numbers (Figures 5-6 and Figures 7-9)
  67. Figure 10 and 11, change x-axis title to ‘Number of cycles to failure, Nf
  68. Line 204 and 205: change the caption to: S-N data of ... material in nominal condition (...temperature, in air) and samples immersed in .... solution for 7 and 14 days prior to bending fatigue tests.
  69. The data points should be added on the graphs for all samples in fig 10 and 11.
  70. Figure 11: delete the minor grids for y-axis.
  71. Add another figure to show both test series (batches) in different immersion duration on the same graph and compare.
  72. Line 210, modify ‘bent samples’ to ‘under cyclic bending load’.
  73. Line 215, Specify the Pearson correlation coefficient for each graph separately.
  74. Line 215, for second Serie (fig 11) 28 days of pre corrosion is not different with 7 and 14 fays, it is even higher. So this sentence should be modified.
  75. Line 216, modify ‘the length of immersion time’ to ‘immersion duration’
  76. Line 218, Clarify: have you observed the pits on the samples before fatigue test, what was the size of pits and the number of pits on the surface? You need to prove ‘pits become deeper’ by observed images or by pit depths.
  77. Line 224, clarify this sentence.
  78. Line 230. What do you mean by ‘and the greater they are,’? The writing of this sentence should be improved.
  79. Line 234, rapid fracture region looks the same in fig 3 a and b! Modify this sentence.
  80. Line 235, elaborate on what made you result on the increased rate of plastic strain
  81. Line 240, close up view on crack origin is required to see whether the crack is initiated from corrosion pits.
  82. Line 243, this can happen for fatigue tests in a corrosive environment, not for corroded samples. Please elaborate.
  83. Line 242 to 247 are not relevant to this study.
  1. A ‘Conclusions’ section should be added after the discussion section.

Author Response

Dear Reviewer.

Thank you for all the remarks concerning the article. Detailed information on the changes can be found below. Due to the number of comments, some of the responses here are grouped. It has been re-edited following also the remarks by other reviewers. We sincerely hope that now it is written in a convenient and proper manner.

Comments 1-2:

The suggested changes have been applied.

Comment 3:

“Line 17 to 19 should be modified to represent obtained results.”

Response:

The lines have been changed. Before:

“The characteristic curves show the susceptibility of the test material to corrosive processes, which results in a decrease in fatigue life on a larger scale along with the time of pre-corrosion.”, and after:

“The characteristic curves show the susceptibility of the test material to corrosive processes, which results in a decrease in fatigue life along with the increase of pre-corrosion time. The samples with longer immersion duration revealed larger surface losses and widespread of corrosion pits.”

Comment 4:

“Line 23, this parameter is not used in the paper.”

Response:

The parameter is used in Table 5 and in line 193.

Comment 5:

“Line 24, delete ‘elastic’ and use ‘applied’ instead.”

Response:

Assumed it should be line 24 instead of 25, line 25 is now changed.

Comment 6-11:

The suggested changes have been applied.

Comment 12:

“Line 40, specify ‘others’”

Response:

This part was referring to military applications, it is now clarified.

Comment 13:

“The English of the introduction should be improved. It is not easy to read and clear.”

Response:

Introduction is now re-edited, mostly according to the following comments. The changes are visible in the manuscript, as well as in the following responses.

Comment 14:

“More references should be added to the introduction to identify the importance of the problem investigated in this issue and also to highlight the gap in the literature.”

Response:

More references have been added that identify the importance.

The gap in the literature can be found e.g. by a number of references to different loading types ([12-19] in the paper) and almost no reference to bending.

Comment 15:

“Line 42 and 43, modify ‘ wide possibilities of joining in processes such as melting (welding) or friction stir welding (FSW, including with other materials)’ to ‘weldability’.”

Response:

The authors wanted to emphasize the possibilities of joining with other materials Therefore, realizing the complexity of the sentence, it was changed in a different way.

The lines have been changed into:

“good ability to be welded, including with other materials”

Comment 16:

“Line 43 to 45, Sentence ‘These properties make it possible to reduce operating expenses, such as operating costs (low weight of the elements) and maintenance costs (corrosion resistance higher than that of ferrous metals).’ should be modified, it is not easy to read. Operating expense is the same as operating cost!”

Response:

The lines have been changed into:

“These properties make it possible to reduce operating costs: low weight  means low energy consumption. In the same way, maintenance costs can also be reduced: corrosion resistance higher than that of ferrous metals allows to plan less maintenance effort.”

Comment 17:

“Line 46, sentence ‘Corrosion processes are the cause of degradation of the material as a result of the destruction processes that lead to a decrease in strength and operational parameters.’ should be modified, it is not good in English and ‘operational parameter’ should be specified.”

Response:

The lines have been changed. Before:

“Corrosion processes are the cause of degradation of the material as a result of the destruction processes that lead to a decrease in strength and operational parameters.” and after:

“Corrosion processes are the cause of degradation of the material. They arise as a result of the destruction processes and lead to a decrease in strength and operational parameters (i.e. fatigue life).”

Comment 18:

“Line 47, corrosion pit should be defined first.”

Response:

The lines have been changed. Before:

“The formation of possible corrosion pits in the material will result in a loss of strength due to a source of stress concentration. The initiation of such a process has its origin in the chemical or electrochemical environmental impact.” and after:

“The initiation of such a process has its origin in the chemical or electrochemical environmental impact. In electrochemical corrosion, a typical structure damage is initiated by a corrosion pit, small hole created in a process of localized corrosion. The formation of possible corrosion pits in the material will result in a loss of strength due to a source of stress concentration.”

Comment 19:

“Line 50, should be modified. The electrolyte is one of the necessary elements for any corrosion cell. The presence of cathodes and anodes are important too.”

Response:

A new line has been added: “In such environment local depassivation occurs on a small area of the metal which becomes an anode while the nearby area behaves as a cathode.”

Comments 20-23:

The suggested changes have been applied.

Comment 24:

“Line 64 and 65 should be improved in writing.”

Response:

The lines have been changed. Before:

“Relatively few of them concern bending fatigue or multiaxial fatigue, and these are mainly in specialist applications” and after:

“Relatively few of them concern bending or multiaxial conditions, and these are mainly in specialized applications.”

Comment 25:

“It is not clear why bending fatigue is investigated. Specify some real-life application for this material that is under bending fatigue. The same for used corrosive environment. Why is this corrosive environment chosen? Is it replicating any in-service condition? These all should be described in the introduction.”

Response:

Bending, as well as tension/compression and torsion is one of the basic loading types. As it was already stated, state of the art still lacks basic data on the fatigue behaviour of a such degraded aluminium alloy and this is what inspired this research manuscript.

3.5 wt % NaCl solution in water, was chosen in order for the experiment to be reproducible. The related lines are  96 and 97: “A 3.5% salinity was chosen as a reproducible solution for tests on corrosion, recognised by the ASTM D1141-98 standard for the preparation of saline water.” Also, lines 68 and 69 in introduction were changed. Before:

“The material tested was aluminium alloy AW-2017A-T4 after various periods of exposure to conditions favourable to electrochemical corrosion.” and after:

“The material tested was aluminium alloy AW-2017A-T4 after various periods of exposure to 3.5 wt % NaCl solution in water, i.e. conditions favourable to electrochemical corrosion recognised by the ASTM D1141-98.”

Comment 26:

“Line 75 and 76 should be moved to introduction”

Response:

The suggested changes has been applied.

Comment 27:

“Line 80 says this alloy is difficult to weld whereas in line 42 and 42 the opposite is said! The same for being corrosion resistant.”

Response:

This line was changed. Before:

“Due to the presence of copper in the chemical composition, this alloy is classified as difficult to weld and moderately 80 corrosion-resistant”, and after:

“Due to the presence of copper in the chemical composition, among other alloys it is classified as difficult to weld and moderately corrosion-resistant”.

Comment 28:

“Which specific standard was used to design the samples?”

Response:

The samples used in test are diabolo-shaped specimens. The related line is line 83: “The samples had the geometry of a diabolo specimen”.

Comment 29:

“Which surface preparation process was carried after cutting the samples? What was the sample surface roughness before and after corrosion?”

Response:

The specimens were manufactured in a standard machining process with finishing turning at the end, surface roughness Ra before the corrosion was 1.25. Information was added in line 84: “Surface roughness Ra before the corrosion was 1.25”.

Roughness after the corrosion was not measured.

Comment 30:

“The methodology of corroding the samples should be described in detail. Specify any electrodes used, voltage, etc. Add images of the corrosion cell with the samples.”

Response:

Corrosion process was thoroughly described in subsection 2.2: “Corrosion process”.  It was initiated by immersion in a 3.5 wt % NaCl solution in water (no voltage etc.).

Comment 31:

“Add image of fatigue test machine with the sample inside it."

Response:

A new Figure – Figure 3 is now added.

Comment 32:

“The number of samples, duration of the corrosion process and applied stress should be specified here"

Response:

No place was specified, but all of the above can be found in subsections 2.1, 2.2 and 2.3 – “Material and Samples”, “Corrosion process” and “Fatigue Tests”, respectively.

 Comments 33-34:

“The number of samples, duration of the corrosion process and applied stress should be specified here"

These are references, figure captions are now modified.

Comment 35:

“Line 93, add a unit to the caption (mm)"

Response:

Corrected.

Comment 36:

“Line 96, specify ‘several days’"

Response:

Corrected, now this line is: “The corrosion process of the samples was initiated by immersion in an electrolyte, a 3.5 wt % NaCl solution in water, for 7, 14 or 28 days (depending on the test series)

Comment 37:

“Line 99, clarify whether you mean tap water by ‘running water’ or you mean electrolyte was running in the cell through an inlet/outlet"

Response:

Tap water, the manuscript is now re-edited.

Comment 39:

“Clarify why 7, 14 and 28 days were chosen."

Response:

These durations were chosen in order for the results to be easier to interpret. The 7 and 14 days durations are common in tension/compression tests, 28 days are only for the demineralised water series. It was chosen in order to document the slight influence of pre-corrosion time on fatigue life.

Comment 40:

“It is not clear which samples, how many days. Add a table and detail number of days, sample series and type of electrolyte."

Response:

New Table 2 is now a part of the manuscript.

Comment 41:

“Line 104 to 106, do you mean the same solution was used for all samples and it was not refreshed anytime you put new sample?? Clarify this. Fresh solution should have been used for each sample series.”

Response:

The same solution was used for all samples immersed in a given electrolyte - all of the samples were immersed at the same moment, the difference was that some of them were taken out after 7 days and some after 14 days (or 28 days - in case of demineralised water solution). These lines were corrected in the following manner:

“The samples of a given series were immersed at the same time and in the same container from which all samples of the variant were removed after an appropriate period of time (i.e. after 7, 14 or 28 days, depending on the variant), then washed with demineralized water, and dried.”

Comment 42:

“What was room temperature?”

Response:

About 20 °C, which is in agreement with the definition according to the Oxford English Dictionary.

Comment 43:

“Table 2, add chemical composition of water for the first batch of samples to this table. It should be added to enable the reader to compare the two solutions.”

Response:

This table already shows chemical composition of water for the first batch of samples. The second batch was prepared with the application of demineralised water.

Comment 44:

“Was fatigue tests load controlled or strain-controlled?”

Response:

They were load controlled (line 124). One of the last reviewers had a similar remark:

“Do the fatigue test follow any Standard? Please, specify. Are they strain-controlled? If the authors say that tests are performed at constant-amplitude load, how can the failure criterion be a bending moment drop of 15%?”

Response:

Data evaluation was performed according to Practice for Statistical Analysis of Linear or Linearized Stress-Life (S-N) and Strain-Life (-N) Fatigue Data (ASTM International 2004).

The test result was interpreted as a number of cycles very close to the rupture, but not exactly it. Due to the simplicity of mechanical displacement limiter and its adjustment a simple emergency cut-off could not be representative as a failure criteria. In the last propagation stage when the crack depth and propagation rate is large enough to impose dynamic motion of the load-lever there is a short time interval with drop of the bending moment (at the very end of test). Number of cycles corresponding to this 15% drop was taken as the failure criteria.

Comment 45:

“Section 2.3. add a figure of the test set up and test bench with the sample inside the fatigue test machine”

Response:

This was the case of the remark in comment 31 - Figure 3 is now added.

Comment 46:

“Line 120, was your fatigue tests rotating bending? Specify what kind of bending load was applied and how.”

Response:

Fatigue test were carried under bending load only, without torsion. Line 118 is now modified. Before:

“samples were subjected to constant-amplitude loads”, and after:

“samples were subjected to constant-amplitude bending loads”.

Comment 47:

“Line 124, assumption of 15% needs to be supported by references.”

Response to this comment is given in the full response to comment 44 above.

Comment 48-49:

The suggested changes have been applied.

Comment 50-51:

“Figure 3 and 4: Tables should be introduced before this figure and then use the sample numbers in the caption. Show the crack origin and crack growth region and fast fracture zones on the images by arrows and brackets. State in the cation which microscope was used and how many days they were immersed.”

Response:

The tables have now been moved. Due to the  editing limitations, these areas were marked through letters and crack origin is described in the figure caption.

Those images were taken by a photo camera, days of immersion can now be found in caption also, for example:

“Fatigue fracture of exemplary samples from a series subjected to the solution in tap water, submitted to stress amplitude σa = 162 MPa: a) 7 days of immersion (Sample No. 3, Table 4.1)”

Comment 52:

“Line 143 to 149, clarify how does corrosion promoted fatigue crack in this case.”

Response:

Comment 53:

“What is the difference of fig 3 and 4?”

Response:

Those are two different series, they are differing in solution used as it can be seen in figure captions.

Comment 54-55:

“Figure 5 and 6, which part of cross-section is these image from? add another image from cross-section to fig 5 and show where images are taken from by red boxes.”

Response:

New figures have been added according to the suggestions.

Comment 56:

“Line 161, clarify what made you result in the plastic strain in the images.”

Response:

New line is now added: “Plastic deformation of the material was initiated in the surface layer in grains with the crystallographic orientation in the easy glide direction and near the stress concentrators (i.e. in weakened parts of the material).”

This part of the manuscript  is now moved into the Discussion section, following the suggestions of a different reviewer.

Comment 57:

“Line 166, Grammar mistake ‘it was be observed that’.”

Response:

Corrected.

Comment 58:

“Line 167, show pits with an arrow on fig 5a to 9a. ‘÷’ modify to ‘-‘”

Response:

The arrows are now added according to the suggestions.

Comment 59:

“Line 167, change ‘corrosive’ to ‘corrosion’”

Response:

Corrected.

Comment 60:

“Line 167, what was the corrosion pit depth? Clarify.”

Response:

The more appropriate word is widespread instead of deeper. This change is now applied.

Comment 61:

“Have you observed corrosion pits at the crack origin? Add microscopic images of crack origin showing the pit.”

Response:

The pits were observed on the surface and on the fracture cross section (Fig. 7a, 9a-12a) of tested alloy. New line is added to the Discussion section: “Corrosion pits were observed also at the crack origin.”

Comment 62:

“Line 169, English should be improved.”

Response:

The lines have been changed. Before:

“Such a phenomenon lead to local plastic strain of the 168 material and, consequently, we are dealing with plastic fatigue cracking, which is indicated by ductile 169 fractures of the analyzed specimens”, and after:

“Such a phenomenon leads to local plastic deformation of the material. It points on plastic fatigue cracking, which is indicated by the ductile fractures of the analysed samples”.

This part of the manuscript  is now moved into the Discussion section, following the suggestions of a different reviewer.

Comment 63:

“Line 170, modify ‘÷’ to ‘to’”

Response:

Corrected, assuming that ‘to’ should be ‘-’.

Comment 64:

“Line 165, clarify which pits you mean on the images.”

Response:

The pits are now shown in Figures 9a, 11a-14a.

Comment 65:

“Line 184, delete ‘one’”

Response:

Corrected.

Comment 66:

“Line 201, use the number of tables here instead of the figures numbers (Figures 5-6 and Figures 7-9)”

Response:

Corrected.

Comment 67:

“Figure 10 and 11, change x-axis title to ‘Number of cycles to failure, Nf’”

Response:

Corrected.

Comment 68:

“Line 204 and 205: change the caption to: S-N data of ... material in nominal condition (...temperature, in air) and samples immersed in .... solution for 7 and 14 days prior to bending fatigue tests..”

Response:

Corrected, Figure 11 caption was also corrected.

Comment 69:

“The data points should be added on the graphs for all samples in fig 10 and 11”

Response:

Coordinates of the points corresponding to those S-N curves can be found in Tables and data repository.

We appreciate the comment and understand the intentions. Additional figures for each table of results were originally in the text, but the manuscript was once rejected because of they were there. One of the reviewers insisted on deleting them. Below you can find the correspondence:

“(…) now it is not clearly what you compare and why you have these figures in this article...(the figures 4 to 7 are not necessary - better are the fig. 8 and 9, these are ok)”

And the remark from the next round of review:

“The contribution still contains Figures 5 to 9. These are duplicates, because Figures 10 and 11 describe the same results. But better are figures 10 and 11 for explanation the fatigue properties!”

Response:

In the authors' opinion Wohler diagrams with experimental points for each series of test are necessary in order to accurately show the test results and validate the regression analysis.

These figures were generated because they show the location of experimental points in regards to the regression line. In this way they validate the regression results.

Comment 70:

“Figure 11: delete the minor grids for y-axis.”

Response:

Minor grids are now deleted.

Comment 71:

“Add another figure to show both test series (batches) in different immersion duration on the same graph and compare.”

Response:

Showing the curves for all five tests on one figure significantly decreases the readability. Separate figures were already in the manuscript, but they had to be deleted (see response to comment 69).

Comment 72:

“Line 210, modify ‘bent samples’ to ‘under cyclic bending load’”

Response:

Corrected.

Comment 73:

“Line 215, Specify the Pearson correlation coefficient for each graph separately.”

Response:

This line is now modified. Before:

“The obtained test results feature Pearson correlation coefficient values not less than -0.88.”, and after:

“The obtained test results feature Pearson correlation coefficient values between -0.88 and -0.90 in case of tap water solution and between -0.88 and -0.93 in case of demineralised water solution.”

Comment 74:

“Line 215, for second Serie (fig 11) 28 days of pre corrosion is not different with 7 and 14 fays, it is even higher. So this sentence should be modified.”

Response:

This line is now modified. Before:

“The comparison showed a decrease in fatigue strength in each case.”, and after:

“The comparison showed a decrease in fatigue strength in each case except the 28 days of pre-corrosion serie. In this case it was slightly higher than the 7 and 14 days serie, but it has to be mentioned that for this solution all S-N curves are very close to each other.”

Comment 75:

“Line 216, modify ‘the length of immersion time’ to ‘immersion duration’”

Response:

Corrected.

Comment 76:

“Line 218, Clarify: have you observed the pits on the samples before fatigue test, what was the size of pits and the number of pits on the surface? You need to prove ‘pits become deeper’ by observed images or by pit depths.”

Response:

Small pits were visible before the fatigue test. Growth of corrosion pits becomes visible in view of Figures 15-16. Additionally, new lines have been added to the Results as well as the Discussion section:

In the latter:

“The microscopic observations show that all samples of the tested alloy have undergone pitting corrosion. The smallest number of pits was formed on the sample after the shortest immersion duration (7 days, Figures 15a, 16a), more pits were visible on the samples immersed for 14 days (Figure 15b, 16b), which is consistent with the results obtained from fatigue tests. The observations indicate a disruption of the anode layer and the occurrence of micro cracks which transform into corrosion. In addition, it is not possible to create a passive layer on the surface of the material and for this reason it is easier to initiate pitting on its surface. Larger corrosion damage probably results from the extension of the time of exposure to aggressive corrosive environment, which simultaneously translates into shorter fatigue life of the material. The results have shown that after a longer period of exposure to NaCl solution pits are formed on the surface of the samples with higher density.”

Comment 77:

“Line 224, clarify this sentence.”

Response:

At high number of cycles (i.e. low stress values) S-N curves for both pre-corroded samples are very close. It can be seen in view of Figure 10  at approx. 1 x 107 cycles. From the above it can be concluded that at low stresses duration of pre-corrosion (i.e. material structure after corrosion) has smaller influence on fatigue life than at high stresses. The lines are now changed. Before:

“The above observation indicates a lesser influence of material properties on its fatigue strength at low stress values.”, and after:

“The above observation indicates a lesser influence of material properties on its fatigue strength at low stress values and high number of cycles. In this region curves for both immersion durations are very close to each other and the obtained life would be close for both of them”.

Comment 78:

“Line 230. What do you mean by ‘and the greater they are,’? The writing of this sentence should be improved.”

Response:

This line and the line preceding it was related to the discussion on the mechanism of  formation of plastic strains based on the results by Yang et al. The sentence may be confusing, now it is changed. Hope it is more understandable now. Before:

“At higher stress amplitudes, the presence of corrosion pits lead to local plastic strain and the coalescence of flaws, and the greater they are, the deeper the cavity was, which results from the time of exposure to the corrosive environment.”, and after:

“At higher stress amplitudes, the presence of corrosion pits lead to local plastic strain and the coalescence of flaws.  The higher the amplitudes, the deeper the cavity is.”

Comment 79:

“Line 234, rapid fracture region looks the same in fig 3 a and b! Modify this sentence.”

Response:

Corrected.

Comment 80:

“Line 235, elaborate on what made you result on the increased rate of plastic strain.”

Response:

New line is now added (it results also from comment no. 56): “Plastic deformation of the material was initiated in the surface layer in grains with the crystallographic orientation in the easy glide direction and near the stress concentrators (i.e. in weakened parts of the material).”

This part of the manuscript  is now moved into the Discussion section, following the suggestions of a different reviewer.

Comment 81:

“Line 240, close up view on crack origin is required to see whether the crack is initiated from corrosion pits.”

Response:

New figures are now added: Figures 6 and 7.

Comment 82:

“Line 243, this can happen for fatigue tests in a corrosive environment, not for corroded samples. Please elaborate.”

Response:

It is now deleted according to the suggestion below.

Comment 83:

“Line 242 to 247 are not relevant to this study.”

Response:

The lines are deleted now.

Comment 84:

“A ‘Conclusions’ section should be added after the discussion section.”

Response:

A new section is now added.

Yours sincerely,

                                                                                                                                                                                                                                     Łukasz BLACHA,

                                                                                        Joanna MAŁECKA,    

                                                                                        Tadeusz ŁAGODA

Round 2

Reviewer 1 Report

It may be accepted for publication.

Author Response

Thank you for your comments. Now the paper is better.

Reviewer 2 Report

  1. (Line 179, Microscopic images of crack origin showing the pit are shown in Figure 6 and 7) Where are the pits? Please indicate. Also, the resolution of Figure 7 (c) should be improved.
  2. (Line 204, Figure 9 (a)~Figure 14(a)) I am not convinced that this is “pit”. The initiation of pit formed during pre-corrosion procedure should locate on the “surface” of metallic substrate, not the surface of created during fatigue crack propagation. Also, how the pits are “corrosive”?
  3. (Line 219, Figure 15) The resolution of Figure 15 should be improved.

Author Response

Dear Reviewer

  • Line 179, Microscopic images of crack origin showing the pit are shown in Figure 6 and 7) Where are the pits? Please indicate. Also, the resolution of Figure 7 (c) should be improved.

The microscopic images shown in Fig 6 and 7 showing propagation of crack, but not the pits.
The resolution of Fig. 7 was  improved.

  • Line 204, Figure 9 (a)~Figure 14(a)) I am not convinced that this is “pit”. The initiation of pit formed during pre-corrosion procedure should locate on the “surface” of metallic substrate, not the surface of created during fatigue crack propagation.

In our opinion, pitting starts on the surface of the material, but also quickly progresses inwards. Very often the rest of the metal on the surface remains intact.  Such a phenomenon usually occurs in materials that are subject to self-passivation, such as stainless steel or aluminium alloys.  Usually, pits are first nucleated on the metal surface and then develop and propagate inwards, hence the visible pits at material fractures. Pit nucleation occurs at the weakest points of the passive layer: in places of mechanical damage, near some non-metallic inclusions or at grain boundaries. Probably, the nucleation is preceded by adsorption of aggressive ions, especially Cl, on the surface. Then the ions penetrate the passive layer by migration or penetration.

  • Also, how the pits are “corrosive”?

It was a mistake. The remarks were applied according to the suggestions (description of Fig. 9a-14a)

  • (Line 219, Figure 15) The resolution of Figure 15 should be improved.

The remark was applied according to the suggestions (resolution of Fig. 15 was improved)

 Thank your for all cooments. This comments inproved this paper

Łukasz Blacha

Joanna Małacka

Tadeusz Łagoda

Reviewer 3 Report

The manuscript is improved. Thank you. 

Author Response

Thank you for your comments. Now the paper is better.

This manuscript is a resubmission of an earlier submission. The following is a list of the peer review reports and author responses from that submission.

Round 1

Reviewer 1 Report

The geometry of the samples for fatigue test is according some standards or?

Why was used 3.5 % NaCl solution for corrosion process? Also the days for immersion in electrolyte 7, 14 or 28 it is according some rules or why you chose such days?

The figure 2 it is difficult to make conclusion because in this picture you compare the three different samples (the stress, the number of cycles). It will be better if you compare samples immersion in electrolyte for 7 days but with different number of cycles – for declaring the changes in fatigue fracture. Or you compare samples after 7 and 14 and 28 days with the some stress but different number of cycles...

The Wohler diagrams (figures 4 to 7) are not clearly comparable. Better is when you give the two curves to the one diagram. It depends what you would like to compare. For example: you can give together the curve for 7 days in demineralized water and 14 days  in demineralized water....now it is not clearly what you compare and why you have these figures in this article...(the figures 4 to 7 are not necessary - better are the fig. 8 and 9, these are ok)

Author Response

  • Comment 1:

“The geometry of the samples for fatigue test is according some standards or?”

Response:

The samples used in test are diabolo-shaped specimens.

Change:

Line 87 has been changed in order to include more details.

Before:

“The geometry of the samples (Figure 1) was obtained in the processes of sawing and then turning and milling.”

After:

“The samples had the geometry of a diabolo specimen (Figure 1). They were manufactured in the processes of sawing and then turning and milling.”

  • Comment 2:

“Why was used 3.5 % NaCl solution for corrosion process? Also the days for immersion in electrolyte 7, 14 or 28 it is according some rules or why you chose such days?”

Response:

3.5% solution is the most common corrosive agent, it is recognised as a standard for sea water and popular among research labs for tests on corrosion.

Change:

New text has been added in line 97:

“3.5% salinity was chosen as reproducible solution for tests on corrosion, recognised by ASTM D1141-98 standard for preparation of saline water.”

as well as in line 103:

“Such long immersion periods were chosen in order to fully test the influence of pre-corrosion time on fatigue life since many tests are accelerated tests (such as salt spray)

  • Comment 3:

The figure 2 it is difficult to make conclusion because in this picture you compare the three different samples (the stress, the number of cycles). It will be better if you compare samples immersion in electrolyte for 7 days but with different number of cycles – for declaring the changes in fatigue fracture. Or you compare samples after 7 and 14 and 28 days with the some stress but different number of cycles...”

Response:

Figure 2 has been changed, now it shows fracture planes of samples at similar stress amplitudes (7 and 14 days of immersion). Additionally, Figure 3 is now added in order to show fracture planes of samples at similar stress amplitudes (7, 14 and 21 days of immersion in the solution in demineralised water).

  • Comment 4:

“The Wohler diagrams (figures 4 to 7) are not clearly comparable. Better is when you give the two curves to the one diagram. It depends what you would like to compare. For example: you can give together the curve for 7 days in demineralized water and 14 days in demineralized water....now it is not clearly what you compare and why you have these figures in this article...(the figures 4 to 7 are not necessary - better are the fig. 8 and 9, these are ok)”

Response:

In the authors' opinion Wohler diagrams with experimental points for each series of test are necessary in order to accurately show the test results and validate the regression analysis. The curves for batch subjected to the solution in running water are compared together on (old) Figure 8. The same situation applies for batch subjected to the solution in demineralised water - curves are plotted together on (old) Figure 9.

Reviewer 2 Report

Dear Authors

       The authors have tested the bending fatigue of aluminum alloy after its exposure to corrosive medium (3.5% NaCl). The study is interesting and will add some useful data to the scientific literature. However, the study lacks depth and clarity. For example the authors have several times mentioned why this study is important which seems to bore the reader. There several tables which could be converted to interesting figures which will improve the visibility of the data to the reader. I would recommend that the authors revisit this paper and improve the readability. 

Thank you

Author Response

  • Comment 1:

the authors have several times mentioned why this study is important which seems to bore the reader.”

Response:

Several lines have been deleted, it can be seen in the .doc file (“track changes” option).

Change:

Line 45:

“These properties also enable the use of aluminum alloys as a material for construction elements designed to work in a corrosive environment.” – deleted.

Line 47:

The sentence originally on  line 61:

“The formation of possible corrosion pits in the material will result in a loss of strength due to a source of stress concentration.”

has been moved here.

Line 48:

“Corrosion resistance is determined by the material's susceptibility to corrosive processes.” – deleted.

Line 51:

“Chemical corrosion occurs in an environment without ionic conductivity, while” – deleted.

  • Comment 2:

“There several tables which could be converted to interesting figures which will improve the visibility of the data to the reader.”

Response:

Table 1 has been replaced by a bar graph shown in new Figure 1.

The data in tables 4.1-2 and 5.1-3 are already graphically presented in Figures 3-7. The tables are shown in the article text in order to provide the reader with comprehensive results in the most precise data form.

Additionally, if the Editor wishes so, the authors agree for the Editor to delete these tables from the article since they already are stored in a data repository at http://dx.doi.org/10.17632/dt33kwy49z.1,  mentioned in the article text.

Reviewer 3 Report

The authors present a potentially interesting work. Measuring corrosion effects on fatigue life is always important. However, the paper is too simple to fulfill scientific standards.

Mayor concern: it merely present experimental results, but there is no enough explanation. Analysis of the defects caused by corrosion processes, and its relation with the experimental results, should be mandatory.

Other comments:

Abstract, line 17: correction should be corrosion, shouldn´t it?

Introduction deals with some concepts, such as chemical corrosion, electrochemical corrosion, fatigue, etc, that are simplified. This is good, but some sentences look naïve. I would recommend a stricter wording all along the introduction.

Does the geometry of the specimen follow any Standard? Please, specify.

Do the fatigue test follow any Standard? Please, specify. Are they strain-controlled? If the authors say that tests are performed at constant-amplitude load, how can the failure criterion be a bending moment drop of 15%? If you control the load or the moment, you can measure variation on displacements, and viceversa, but in load control (if bending moment variation is constant) you cannot measure drops on it.

Figure 2: please, point out initiation points and crack front at failure.

English requires revision. The sentences are sometimes too vague and there are typos.

Author Response

  • Comment 1:

Mayor concern: it merely present experimental results, but there is no enough explanation. Analysis of the defects caused by corrosion processes, and its relation with the experimental results, should be mandatory.”

Response:

Section “Discussion” is now extended,  added text data can be seen in the .doc file (“track changes” option).

  • Comment 2:

“Abstract, line 17: correction should be corrosion, shouldn´t it?”

Response:

Corrected.

  • Comment 3:

“Introduction deals with some concepts, such as chemical corrosion, electrochemical corrosion, fatigue, etc, that are simplified. This is good, but some sentences look naïve. I would recommend a stricter wording all along the introduction.”

Response:

Introduction is already re-edited as it was also pointed out by the first reviewer.

  • Comment 4:

“Does the geometry of the specimen follow any Standard? Please, specify.”

Response:

Yes, it is a diabolo specimen. Details are in the response to the first reviewer.

  • Comment 5:

“Do the fatigue test follow any Standard? Please, specify. Are they strain-controlled? If the authors say that tests are performed at constant-amplitude load, how can the failure criterion be a bending moment drop of 15%?”

Response:

Data evaluation was performed according to Practice for Statistical Analysis of Linear or Linearized Stress-Life (S-N) and Strain-Life (-N) Fatigue Data (ASTM International 2004).

The test result was interpreted as a number of cycles very close to the rupture, but not exactly it. Due to the simplicity of mechanical displacement limiter and its adjustment a simple emergency cut-off could not be representative as a failure criteria. In the last propagation stage when the crack depth and propagation rate is large enough to impose dynamic motion of the load-lever there is a short time interval with drop of the bending moment, at the very end of test. Number of cycles corresponding to this 15% drop was taken as the failure criteria.

Change:

Line 121:

Before:

“The failure criterion was assumed to be a 15% decrease in bending moment, the corresponding number of cycles was assumed as the durability, Nf.”

After:

“The failure criterion was assumed to be a 15% decrease in bending moment, appearing always at the very end of test. The corresponding number of cycles was assumed as the durability, Nf.”

Line 144:

Before:

“The observed fatigue strength allowed to formulate the fatigue characteristics of type S-N in a double logarithmic system:”

After:

“The observed fatigue strength allowed to formulate the S-N fatigue characteristics in a double logarithmic system (according to ASTM [26]):”

  • Comment 6:

“please, point out initiation points and crack front at failure.”

Response:

The direction of crack propagation has been indicated in the caption of corresponding figures.

Round 2

Reviewer 1 Report

The contribution still contains Figures 5 to 9. These are duplicates, because Figures 10 and 11 describe the same results. But better are figures 10 and 11 for explanation the fatigue properties!

The authors also written in the conclusions: "It can be explained through the fact that 205 with increasing of immersion time, the depth of corrosion pits becomes bigger and fatigue crack 206 initiation life becomes shorter, although the trend decreases in the high cycle fatigue region. These 207 results can be interpreted as an evidence of the importance of the joint effect of stress and corrosion.. "   But it is not possible to find out, from the given article, how they came to these results. Fatigue fracture surfaces (figure 3) indicate only an overall view of the fatigue fracture surface.

I suggest that the space occupied by Figures 5 to 9 be used for a more detailed description and analysis of fatigue fracture surfaces. (Detail of initiation paces, detail on to corrosion attack on fracture surfaces..and so on).

I recemended the citation of the authors and article in this sentence: " The processed data required to reproduce these findings 170 are available to download from http://dx.doi.org/10.17632/dt33kwy49z.1" not the web page.

Author Response

Dear editor, dear reviewer.

Thank you for all remarks concerning the article. Detailed information on the changes can be found below. We sincerely hope, that now it is written in a convenient manner.

  • Comment 1:

“The contribution still contains Figures 5 to 9. These are duplicates, because Figures 10 and 11 describe the same results. But better are figures 10 and 11 for explanation the fatigue properties!”

Response:

These figures were generated because they show the location of experimental points in regards to the regression line. In this way they validate the regression results. Although we agree that it can be too much and the suggested figures have been deleted.

  • Comment 2:

“The authors also written in the conclusions: "It can be explained through the fact that 205 with increasing of immersion time, the depth of corrosion pits becomes bigger and fatigue crack 206 initiation life becomes shorter (…)" But it is not possible to find out, from the given article, how they came to these results. Fatigue fracture surfaces (figure 3) indicate only an overall view of the fatigue fracture surface. I suggest that the space occupied by Figures 5 to 9 be used for a more detailed description and analysis of fatigue fracture surfaces. (Detail of initiation paces, detail on to corrosion attack on fracture surfaces..and so on).”

Response:

The visible decrease of  fatigue lives within the LCF region was attributed to notch effect, reflected in the enlarged crack propagation region of fracture surface. The character of such effect is assumed to be mostly geometrical with a possible slight influence of the material.

Regarding the comment, the “Discussion” section now features a new contribution. Due to the size, the text data was not pasted here although it can be easily seen in the document (“track changes” option is on).

  • Comment 3:

“I recemended the citation of the authors and article in this sentence: " The processed data required to reproduce these findings 170 are available to download from http://dx.doi.org/10.17632/dt33kwy49z.1" not the web page.”

Response:

Changes were made.

Before:

“The processed data required to reproduce these findings 170 are available to download from http://dx.doi.org/10.17632/dt33kwy49z.1”

After:

“The processed data required to reproduce these findings can be found at [27].”

Yours sincerely,

                                                                                                                                                                       Łukasz BLACHA, Tadeusz ŁAGODA

Reviewer 2 Report

I would like to commend the authors for revising the manuscript as per the review recommendation. 

Author Response

Dear editor, dear reviewer.

Thank you for all remarks concerning the article. Detailed information on the changes can be found below. We sincerely hope, that now it is written in a convenient manner.

Comment:

“I would like to commend the authors for revising the manuscript as per the review recommendation.”

Response:

The newly revised version includes more detailed description and analysis of the results and on the defects generated by the different conditions and how they affect the fatigue lives. The “Discussion” section now features a new contribution. Due to the size, the text data was not pasted here although it can be easily seen in the document (“track changes” option is on).

In order to improve the readability, the figures with individual S-N curves were deleted, also following the strong suggestions of another reviewer. The S-N curves were left only in figures 10 and 11, as they better explain the fatigue properties. This may improve the visibility by focusing on the main goal of the research – comparison between bending fatigue lives corresponding to different corrosion conditions.

In order to improve the English language and style, the manuscript has been submitted to MDPI English editing service.

Yours sincerely,                                                                                                                                                                                                                                  Łukasz BLACHA,  Tadeusz ŁAGODA

Reviewer 3 Report

The authors have included some slight corrections. However, the main concern still persists: there is no analysis about the defects generated by the different conditions, and how they affect the fatigue lives. This is crucial. Defects should be characterised, and the relation between such defects and the fatigue performance should be established.

I cannot recomment publication.

Author Response

Dear editor, dear reviewer.

Thank you for all remarks concerning the article. Detailed information on the changes can be found below. We sincerely hope, that now it is written in a convenient manner.

Comment:

“there is no analysis about the defects generated by the different conditions, and how they affect the fatigue lives. This is crucial. Defects should be characterised, and the relation between such defects and the fatigue performance should be established.”

Response:

In order to improve the English language and style, the manuscript has been submitted to MDPI English editing service.

The newly revised version includes more detailed description and analysis of the results with particular emphasis on their relation to fatigue lives. The analysis focused on differences between LCF lives obtained during tests on samples of different immersion times and, additionally,  the proximity of S-N curves in the HCF region. The “Discussion” section now features a new contribution. Due to the size, the text data was not pasted here although it can be easily seen in the document (“track changes” option is on).

In order to improve the readability, the figures with individual S-N curves were deleted, also following the strong suggestions of another reviewer. The S-N curves were left only in figures 10 and 11, as they better explain the fatigue properties after different corrosion conditions.

Yours sincerely,                                                                                                                                                                                                                                  Łukasz BLACHA,   Tadeusz ŁAGODA

Round 3

Reviewer 1 Report

Thank you for acceptance of my recommendation.

Reviewer 3 Report

We are in a circular review. From my point of view, the authors have not addressed the major issue of the paper: there is no characterisation of the defects caused by the different treatments (type? depth? dimensions?), and the corresponding relation with fatigue lives. I cannot recommend publication.